# Survey on *Drosophila suzukii* Natural Short-Term Dispersal Capacities Using the Mark−Release−Recapture Technique

**DOI:** 10.3390/insects10090268

**Published:** 2019-08-24

**Authors:** Sandra Vacas, Jaime Primo, Juan J. Manclús, Ángel Montoya, Vicente Navarro-Llopis

**Affiliations:** 1Centro de Ecología Química Agrícola—Instituto Agroforestal del Mediterráneo, Universitat Politècnica de València–Edificio 6C-Camino de Vera s/n, 46022 Valencia, Spain; 2Centro de Investigación e Innovación en Bioingeniería, Universitat Politècnica de València, Edificio 8E-Camino de Vera s/n, 46022 Valencia, Spain

**Keywords:** spotted-wing drosophila, Diptera, Drosophilidae, dispersion, mark–release–recapture

## Abstract

Spotted wing drosophila, *Drosophila suzukii* Matsumura (Diptera: Drosophilidae), has become a key pest for soft fruits and cherries in Europe in less than a decade since the first outbreak in 2007. Although this pest’s passive dispersal ability has been observed over more than 1400 km in 1 year, active spread has not yet been extensively studied. A mark−release−recapture (MRR) method based on protein-marked flies was employed to determine the flight capacity of *D. suzukii*. Sterile marked flies were released and recaptured in a trap grid at increasing distances from 10 to 250 m from the releasing point to study flight distance during periods ranging from 3 h to 1 week. MRR experiments were replicated in the presence and absence of host fruits to study how they could affect dispersal behavior. The dispersal capacity of the Mediterranean fruit fly, *Ceratitis capitata* Wiedemann (Diptera: Tephritidae) was also studied under the same conditions. The results showed a low dispersal ability for *D. suzukii*, with a daily flight distance below 100 m with no predominant wind. The implications on natural dispersion and control methods based on attractants are discussed.

## 1. Introduction

Spotted-wing drosophila, *Drosophila suzukii* Matsumura (Diptera: Drosophilidae), native to Southeast Asia, has been revealed as a major damaging invasive fruit fly that threatens both the European and American fruit industries [1]. Current *D. suzukii* management relies on insecticides, applied intensively with optimum control by applications every 5–7 days during the fruit ripening season [2,3]. Frequent insecticide treatments can lead to the development of resistance, especially because *D. suzukii* has both high fecundity and generation turnover [4,5,6], besides posing risks to natural enemies and other beneficial arthropods. Research on alternative control methods to be included in Integrated Pest Management programs is crucial to reduce or avoid the aforementioned drawbacks of chemical control. These tactics include cultural management [7,8], biological control with both natural enemies and microbiological agents [9,10,11], and trapping techniques, based mainly on the use of food baits [12,13].

Knowledge on the fly dispersion and dispersal capacity is essential to develop control strategies. Dispersion is defined as the distribution pattern of individuals in a habitat, and dispersal is the ability to spread or distribute from a fixed or constant source [14]. According to historic accounts of distribution and introductions, spotted-wing drosophila has a high potential to disperse and search for suitable areas to live [15]. *D. suzukii* movements from forests and noncrop field margins have also been reported [16]. Thus, the availability of wild noncrop and ornamental alternate hosts adjacent to commercial crops contributes to pest spread and economic impact. Recently, studies of the dispersal ability of *D. suzukii* over extended periods (33–44 days) have demonstrated that *D. suzukii* flies are able to fly up to 9000 m away from the marking point over their entire lifetimes and that seasonal breezes likely facilitate long-distance movement [17].

Marking techniques are frequently used to study the natural movement and distribution of insects in the field. They include mark–release–recapture (MRR), where reared insects are marked in the laboratory, released, and recaptured, and mark–capture experiments, in which wild insects are marked (e.g., by contacting marked plants) and their movements are studied in traps located around the marking point [18]. Specifically, the MRR technique has been widely employed to track the movement of insects by releasing marked individuals and recapturing them at given time and distance intervals after their release. There are plenty of marking procedures for insects, such as the application of paint or ink, fluorescent powders, internal dyes, genetic markers, radioactive isotopes, or, more recently, immunomarking [18]. This last technique consists of marking insects with a protein that can be later be detected in recaptured insects by an enzyme-linked immunosorbent assay (ELISA) [19]. Protein marking offers several advantages over the other methods, for instance, materials are inexpensive; the ELISA analysis is simple, safe, and very sensitive; and vertebrate proteins are reported to be persistent, photostable, heat-tolerant, and water-resistant [18,20].

The objective of this study was to evaluate the short-term dispersal capacity of sterile *D. suzukii* flies using the MRR technique. *Drosophila suzukii* flies were irradiated and immunomarked before being released in citrus orchards. Irradiated *D. suzukii* flies were employed in the experiments to avoid releasing a potentially damaging pest population in the area, even though irradiation may lead to poor fly performance. In order to check the validity of the study with a well-known fruit fly, sterile marked *Ceratitis capitata* Wiedemann (Diptera: Tephritidae) males were also released and recaptured. MRR experiments were conducted during two seasons, and flies were released six times in autumn 2015 and spring 2016.

## 2. Materials and Methods

### 2.1. Mediterranean Fruit Fly Stock Colonies

Sterilized and marked Mediterranean fruit fly pupae (Vienna-8 strain temperature-sensitive lethal) were provided by TRAGSA SA (Valencia, Spain), as part of the local government Mediterranean fruit fly SIT Program (Generalitat Valenciana, Valencia, Spain). Pupae were irradiated under hypoxia using an electron accelerator at a dose of 105 ± 10 Gy and marked with pink fluorescent dye (Day-Glo^®^ Color Corp., Cleveland, OH, USA) [21]. Pupae were transferred to plexiglass cages until adult emergence, and adults were provided with water and sugar until release. The number of released adult males was estimated by sorting and counting the number of empty pupal cases in the emergence trays.

### 2.2. D. suzukii Stock Colonies

*D. suzukii* flies were reared in our facilities at the Universitat Politècnica de València (Valencia, Spain), in a controlled environment chamber at 25 ± 2 °C, 50 ± 5% RH, and a 16:8 (light:dark) photoperiod in plexiglass cages (30 × 30 × 40 cm). Adult flies were fed an artificial diet composed of water, baker’s yeast, sucrose, soy flour, corn flour, ethanol, propionic acid, nipagin, and agar (82.7:5:4.1:0.8:5:0.8:0.4:0.2:0.9 w/w), provided in 90 mm Petri dishes. Eggs were laid on this diet, and larvae also developed in it.

### 2.3. D. suzukii Sterilization and Marking

Groups of 400–500, 3 to 5 day old, *D. suzukii* pupae were packed in sealed plastic bags to remain in a hypoxic atmosphere for at least 5 h before irradiation. Irradiation of flies was carried out at the local government (Generalitat Valenciana) facilities of the Mediterranean fruit fly SIT Program (Caudete de las Fuentes, Valencia, Spain) in a Cobalt-60 irradiation unit (Gammacell 220 Excel; MDS Nordion, Ottawa, ON, Canada). The pupae were exposed to gamma radiation at 40 Gy, as this dose ensures an offspring reduction of over 99%. In a preliminary experiment, flies were irradiated with 10 and 40 Gy, and they were allowed to mate with nonirradiated ones. As a result of the 40 Gy dose, females produced few eggs, only 3.6% of those hatched, and 0.8% finished their development to pupae. The 10 Gy dose had less effect; egg hatching was reduced only to 27.7%, and 43.1% of the emerged larvae still reached pupal stage.

After irradiation, groups of pupae were transferred to plastic cages and provided with water and sugar. Two or three days after emergence, flies were chilled to be marked with a protein. This marking protein consisted of bovine serum albumin (BSA) labelled with a low-molecular-weight compound (CNH hapten) at a protein:hapten molar ratio of 1:19 [22]. Marking was performed by spraying 400 μL of 5 μg/mL BSA–CNH in phosphate-buffered saline (PBS) solution over chilled flies. Flies were allowed to awaken and were provided with water and sugar until release. After sterilization and marking, the control flies were maintained in the laboratory to check that survival was not significantly affected by the procedure.

### 2.4. Study Site and Trap Deployment

This study was conducted in Partida de Benadresa (coordinates: 39°59′41.9″ N, 0°07′20.0″ W), located in the municipality of Castelló de la Plana (Castellón, Spain). The study area had a mixed orchard containing various citrus varieties. An array of 40 traps was used to recapture the released flies. Groups of eight traps following eight different orientations were concentrically deployed at 10, 25, 50, 100, and 250 m from the release point (Figure 1) and covered a total area of ~20 ha.

The traps employed to capture both *D. suzukii* and *C. capitata* were red Drosotraps**^®^** (Biobest Biological Systems SL, Westerlo, Belgium), baited with the corresponding dry lures and a DDVP strip (500 mg dichlorvos; Suterra, Valencia, Spain) as an insecticide to retain insects (Figure 2). The attractant employed for *D. suzukii* was a food bait composed of wine:apple cider vinegar (60:40, v/v) + sugar (20 g/L), contained in 50 mL dispensers (Ecologia y Protección Agrícola SL, Valencia, Spain). Trimedlure plug dispensers (1.4 g load; Aragonesas Agro, Spain) were employed to attract *C. capitata* males. A tailored plastic mesh was fitted at mid-height inside the trap to prevent flies from coming into contact with attractants (Figure 2).

### 2.5. Release of Flies and Sampling

Within three days of eclosion, sterile marked flies were transported to the field study site and were released from a single point at the center of the trap array (Figure 1). To ensure that human movements did not assist fly dispersal, cages were carried from the vehicle to the release point, and flies were removed by hand from the equipment and persons before leaving the release point. Flies were released six times, three in autumn (14 October, 27 October, and 11 November 2015) and three in spring (12 April, 26 April, 17 May 2016), under the most favorable conditions for *D. suzukii* survival in the study area (mild temperature and high relative humidity). Climate data for the period of study (1 October 2015–31 May 2016) were obtained from the Benadresa (Castellón) weather station (Instituto Valenciano de Investigaciones Agrarias (IVIA)), located approximately 400 m away from the orchard (Table 1). Flies were released only when conditions were not windy (mean wind speed below 1.5 m/s or maximum wind speed below 5 m/s). The wind conditions during each trial are shown in Table 1. The emergence of irradiated *D. suzukii* at 40 Gy was checked through triplicate measurements and resulted in an average value of 61.4 (±4.7)% (average ± SE). The *C. capitata* emergence at 105 Gy was 91.5 (±2.6)% (data provided by Generalitat Valenciana).

The traps in the array were inspected at different time intervals: 3, 24, and 48 h after release. Traps were also inspected 1 week after release during the trials performed in spring. In all cases, trap contents were emptied into labeled Petri dishes to be transported back to the laboratory. The captured *C. capitata* were inspected under black light (365 nm) to identify the released Mediterranean fruit flies. Detection of the *D. suzukii* marking protein was performed by sandwich-ELISA tests.

### 2.6. Marking Protein Detection—ELISA

Individual *D. suzukii* flies were transferred to Eppendorf microtubes, washed with 200 μL of washing buffer (PBST, 10 mM phosphate buffered saline containing 0.05% Tween-20), and vortexed for 10 s to obtain the samples submitted to the ELISA tests. Fly samples were maintained at −20 °C until analysis. ELISA plates (Costar^®^, Corning, NY, USA) were coated overnight with 100 µL of monoclonal antibody CNH36 (specific for the CNH hapten, Abad and Montoya 1994) at 5 µg/mL in 50 mM of carbonate buffer at pH 9.6. The unbound antibodies were washed off by rinsing three times with PBST. Next, 100 µL of fly washing buffer was added to each well, and plates were incubated at room temperature for 30 min. Each plate included at least one negative and one positive control, where 100 µL PBST and 100 µL sample wash of a recently marked fly were used, respectively. After plate washing, monoclonal antibody CNH36 conjugated to horseradish peroxidase (in accordance with the manufacturer’s instructions, Abcam PLC, Cambridge, UK) was added at 0.5 µg/mL in PBST (100 µL/well), and the plate was incubated for 30 min at room temperature. Then, wells were washed five times using PBST. After washing, substrate solution (2 mg/mL o-phenylenediamine in 25 mM citrate −62 mM phosphate at pH 5.4, containing 0.012% H_2_O_2_) was added (100 µL/well), and the color reaction was developed for 10 min, followed by stopping with an equal volume of 2.5 M H_2_SO_4_. The absorbance of wells was measured at 490 nm in an ELISA plate reader (SpectraMax 190, Molecular Devices, Sunnyvale, CA, USA). The wells developing color contained samples of flies marked with BSA. A detection threshold was set as the mean of the negative samples (10 values of different plates) plus four standard deviations. The wells showing absorbance above this threshold were considered positive, that is, they contained a marked fly.

### 2.7. Data Analysis

The data variability of the total numbers of *D. suzukii* flies and male *C. capitata* captured per time interval were analyzed according to species, distance from the central release point, time passed after fly release, release date, and trap orientation, using a multifactor analysis of variance (MANOVA) [23]. Then, an ANOVA was conducted separately with the data for each species and season. Data were log-transformed prior to the analysis to homogenize variance. Post-hoc Tukey HSD (Honestly Significant Difference) tests were employed for multiple range comparisons (significance indicated by *p* < 0.05).

## 3. Results

As the *D. suzukii* dispersal pattern significantly differed from that of *C. capitata* when the whole data set was included in the MANOVA (factor insect: F = 482.76; df = 1.1663; *p* < 0.001 and factor season: F = 13.52; df = 1.1663; *p* < 0.001), an analysis was performed separately for each insect and also per season. The distance × insect interaction was also significant (F = 15.56; df = 4.1663; *p* < 0.001), indicating that each species has a different dispersal capability. The significant interaction insect × season (F = 12.35; df = 1.1663; *p* < 0.001) indicates that each species behaves differently depending on the season. In general, the recapture value for *C. capitata* was higher than that obtained for *D. suzukii* (Table 1), which was most probably due to the different attractant employed–trimedlure, a powerful parapheromone, for *C. capitata* and the nonspecific food-bait (wine + apple vinegar + sugar) employed for *D. suzukii*.

### 3.1. Recapture of D. suzukii

Overall, during the three release trials performed in October–November 2015, 122 sterile marked *D. suzukii* flies were captured in the trap array and were significantly affected by the time after release and distance but not by orientation from the release point (Table 2). The mean number of released *D. suzukii* was 667 flies, and the distribution of captures according to distance was as follows: 47.5% at 10 m, 13.1% at 25 m, 12.3% at 50 m, 12.3% at 100 m, and 14.8% at 250 m from the release point. Regarding sampling intervals, 2.5% of all the *D. suzukii* flies were recaptured within the first 3 h after release, whereas most were captured 24 h and 48 h after (22.1% and 75.4%, respectively). The distance × time interaction was significant (Table 2), which means that the recapture distance achieved by *D. suzukii* depended on the time since release. Accordingly, 3 h after release, *D. suzukii* were captured at a maximum distance of 25 m, whereas some flies were recaptured at 250 m at 48 h after being released (Figure 3).

During the release trials performed in April–May 2016, 204 sterile marked *D. suzukii* flies were recaptured and were significantly affected by all of the considered factors (Table 2). As the mean number of released *D. suzukii* was 2000 flies, the recapture percentage was even lower than during the autumn trials, as shown in Table 1. The majority of *D. suzukii* were captured at 10 m (77.0%), whereas significantly fewer flies reached further distances (16.2% at 25 m, 5.4% at 50 m, 1.5% at 100 m, and 0.0% at 250 m from the release point). Regarding the time factor, most (46.1%) of the *D. suzukii* flies released in the spring experiments were recaptured within the first 3 h after being released (26.5% 24 h, 13.7% 48 h, and 13.7% one week after release). Once again, the distance × time interaction was significant (Table 2), which means that the recapture distance accomplished by *D. suzukii* depended on the time since release. Accordingly, 3 h after release, *D. suzukii* were captured at a maximum distance of 25 m, whereas some flies were recaptured at 250 m at 48 h after being released (Figure 4).

### 3.2. Recapture of C. capitata

The trap array captured 1161 *C. capitata* during the three autumn releases, and the flies were significantly affected by all of the considered factors (Table 2). The majority of the released medflies were recaptured at 10 m (78.6%), whereas significantly fewer captures were recorded for the other distances (9.0% at 25 m, 5.3% at 50 m, 3.8% at 100 m, and 3.3% at 250 m from the release point). Conversely to *D. suzukii*, 47.6% of the released medflies were recaptured 3 h after being released, whereas 37.5% and 14.9% were captured 24 h and 48 h later, respectively. The distance × time interaction was not significant, as the majority of flies were recaptured at 10 m, regardless of the time since release (Figure 3). However, the orientation × distance interaction was significant; the distance reached depended on orientation, which suggests that flight was probably influenced by the prevailing breeze direction during the MRR experiments.

The total number of *C. capitata* captured during the three spring releases was 2461, with variable recapture percentages (Table 1). The trapping results were significantly affected by the distance from the release point (Table 2; as follows: 54.6% at 10 m, 12.3% at 25 m, 8.4% at 50 m, 11.5% at 100 m, and 13.2% at 250 m) but not by the orientation or time after release (Table 2). On this occasion, the distribution of medfly captures among the sampling times was not significantly different (33.3% at 3 h, 22.8% at 24 h, 20.1% at 48 h, and 23.8% at 1 week after release) (Figure 4). The orientation also showed a nonsignificant effect, but the distance × time interaction was significant (Table 2), which means that the recapture distance achieved by the *C. capitata* sterile males depended on the time since release.

## 4. Discussion

Three MRR experiments were carried out in October–November 2015, and three more in spring 2016, providing recapture data for two seasons with different host densities. *C. capitata* displayed nearly the same behavior for both seasons, but *D. suzukii* behaved differently. In autumn, with ripe and damaged citrus fruits available in the orchard, the *D. suzukii* recapture was almost negligible within the first 3 h after release. However, in spring, when study orchards contained no ripe fruits, large numbers of *D. suzukii* were trapped at 10 m within the first 3 h. This fact can be attributed to the presence of decaying fruits in the experimental field in autumn, as it has been previously reported that fallen split citrus fruit can serve as shelter and a food source for *D. suzukii* [24,25,26]. A short dispersal distance when abundant hosts are available has been reported for other fruit flies of different genera, e.g., *Anastrepha*, *Bactrocera*, and *Ceratitis* [27,28,29]. Plant and Cunningham [30] found a symmetrical dispersal pattern centered on the release point of *C. capitata* in a large macadamia nut orchard. This might be a case where natural hosts are absent, as fly dispersal has been demonstrated as being patchy in a fruit-bearing citrus orchard [14,31].

Regarding hosts, the field study site was homogeneous, with no host other than mid-season clementines within a distance of less than 1.3 km around the release point. Wild blackberries were located in a ravine in the SE direction at 1.3 km, and some backyards lay 1.4 km away in the E direction. However, no influence of these hosts was observed on *D. suzukii* movements. Besides host phenology, climate factors may also play an important role in insect movement [32]. The mean temperature and relative humidity conditions were quite homogenous during both seasons (15.8 °C and 78% in autumn; 16.3 °C and 70.5% in spring) and fell within a suitable range for *D. suzukii* activity [6]. Possible effects of the prevailing wind direction on *C. capitata* dispersal were noticed in some releases, as indicated by the significant effect of the trap orientation on captures. However, it must be taken into account that these experiments were not conducted under windy conditions, and therefore, the results are expected to show the actual fly dispersal capacity when they are not wind-assisted.

*Drosophila melanogaster* Meigen migration towards suitable habitats has been reported from fly release points within 100 m of a desert oasis [33]. *Drosophila obscura* Pomini and *D. subobscura* Collin dispersed up to 100 and 200 m per day, respectively, reaching 310 m per day as a maximum radial distance [34]. In a preliminary MRR study using fluorescent dusts, Lee et al. found that *D. suzukii* moved approximately 67–87 m within 36 h [16]. These studies agree with our results insofar as *D. suzukii* have been shown to hardly reach distances of 100 m within 24 h. However, *D. suzukii* easily fly 25 m within only 3 h after release. This is in accordance with the results obtained by Wong et al. [35], who demonstrated that the median *D. suzukii* flight in a flight mill was 27.16 m. When we focused on dispersal capacity in 1 week after release, we observed that *D. suzukii* mainly remained within the first 50 m from the release point. Using a different attractant, *C. capitata* dispersed along the whole studied 250 m distance. Although we found variability among the six replicates, we highlight that *D. suzukii* were able to reach distances of at least 250 m from the release point within 24 h, albeit in a very small proportion, as the majority of the *D. suzukii* were recaptured within the first 100 m. Compared with *C. capitata*, *D. suzukii* took longer time to fly the same distance, although we should consider that we used a more powerful attractant for *C. capitata*, which could have skewed our observations. This result agrees with a previous study conducted in a cherry orchard, in which the maximum dispersive distance for 95% of the released *D. suzukii* was ca. 90 m [36].

Unlike all *D. melanogaster* group species studied to date, it has been demonstrated that *D. suzukii* does not produce any volatile sex pheromone [37]. As a result, *D. suzukii* attraction relies on inexpensive nonspecific fermented food baits (mainly mixtures of wines and vinegars) and yeast–sugar solutions [12,38,39,40,41], although many efforts have also been made to develop synthetic lures based on fermenting odors [42,43,44,45]. The fact that we used this nonspecific food bait with little attractant power for the MRR experiments might explain the mean low percentage recapture obtained compared with the recapture levels achieved with trimedlure, the powerful parapheromone for *C. capitata* males [46]. Likewise, the attractant efficacy also had an effect on the distance reached by flies. In general, the biggest number of marked *D. suzukii* flies was captured within the first 10 m, and they rarely reached 100–250 m, even after 3 h. Although *C. capitata* were captured near the release point, they were also able to reach traps at 250 m within the first hours after being released.

The immunomarking technique was previously employed to monitor *D. suzukii* activity by Klick et al. [16]. Their protocol was based on spraying a 10% chicken egg white mark solution on field margin vegetation. Then, adult *D. suzukii* were collected from traps and analyzed for the presence of the egg white mark using an egg white-specific ELISA to prove the movement of *D. suzukii* from surrounding vegetation to commercial crops. In our experiments, we sprayed the BSA–CNH marking protein directly on *D. suzukii* flies to allow high sensitivity for marking protein detection on the recaptured flies, even 7 days after being released.

## 5. Conclusions

In conclusion, *D. suzukii* has a low dispersal capacity compared with the distances reported for other fruit flies [46]. We estimate a daily dispersal of 200–300 m for *D. suzukii* when no prevailing wind is present. This conclusion agrees with previous studies in which high *D. suzukii* activity levels were shown to take place within the field margin of cultivated crops surrounded by host plants [16]. Invasion and greater damage were observed in the first rows of cultivated crops near forestry areas, but damage was dramatically reduced in four or five rows (less than 50 m) from the field edge, which demonstrates low daily dispersal ability. However, a recent study showed that *D. suzukii* populations are able to move up to 9000 m in 33 days. That type of long-distance movement was explained by seasonal breezes and updrafts induced by differences in temperature between area elevations [17]. However, in that study the initially marked population was not known and, therefore, it is not possible to determine the percentage of *D. suzukii* that moved such long distance.

Given the low short-term dispersal capacity of *D. suzukii* and the availability of suitable attractants, we recommend a mass trapping strategy using high trap densities, to effectively trap flies at short distances, or perimeter trapping to avoid pest intrusion from field margins that contain alternative hosts or population reservoirs.

## Figures and Tables

**Figure 1 insects-10-00268-f001:**
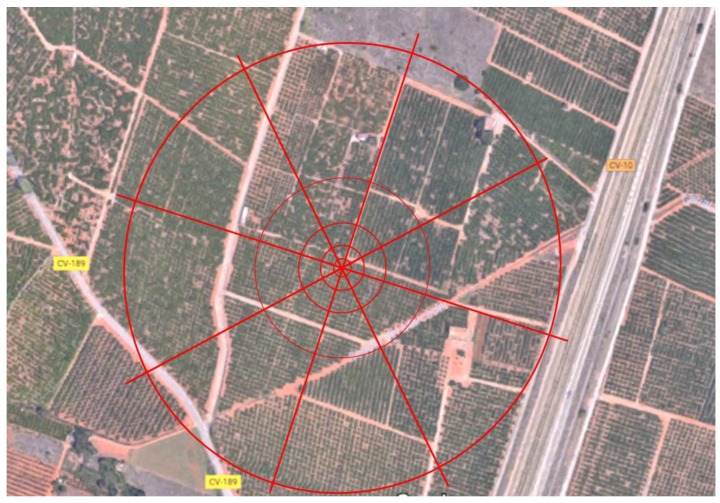
Study area (Partida de Benadresa, Castelló de la Plana, Spain) and distribution of the trap array concentrically at 10, 25, 50, 100, and 250 m from the release point: a trap was placed at each intersection.

**Figure 2 insects-10-00268-f002:**
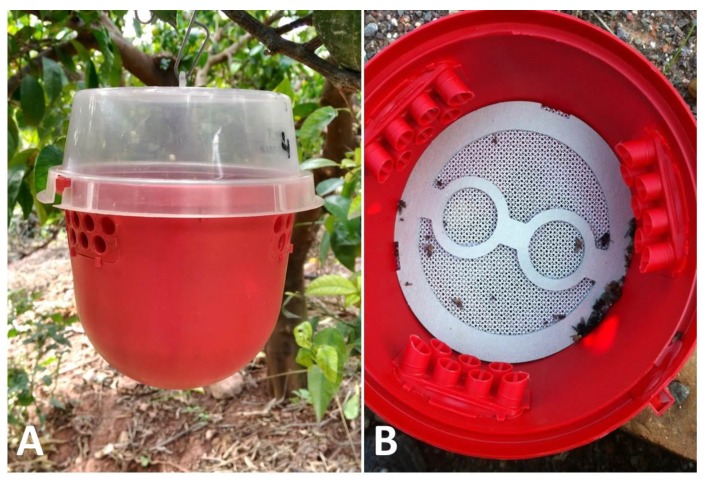
Trap employed to capture both *D. suzukii* and *C. capitata* (**A**), with tailored mesh placed at mid-height to avoid the captured flies coming into direct contact with attractants (**B**).

**Figure 3 insects-10-00268-f003:**
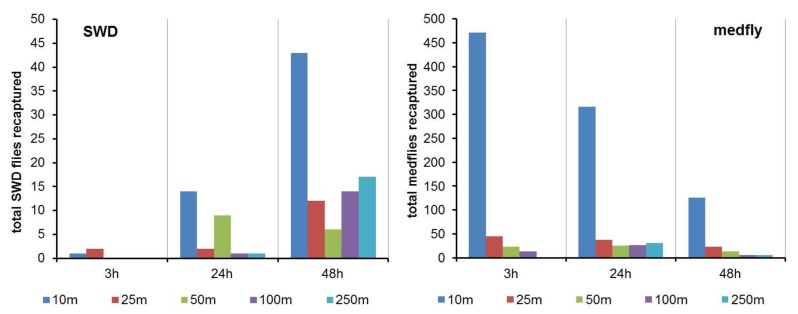
Total number of recaptured flies (left: *D. suzukii* (SWD); right: *C. capitata* (medfly)) at each sampling interval and at different distances from the release point during the three autumn 2015 trials.

**Figure 4 insects-10-00268-f004:**
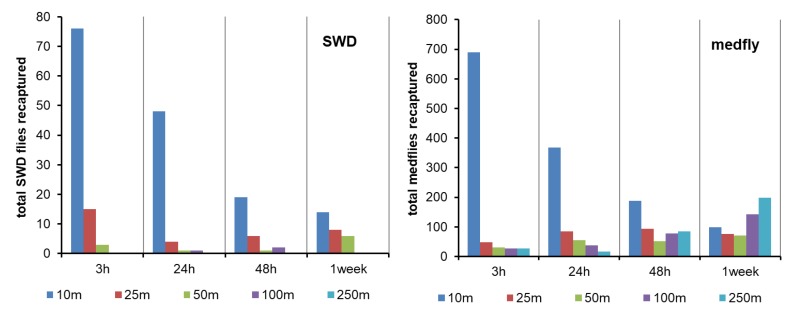
Total number of recaptured flies (left: *D. suzukii* (SWD); right: *C. capitata* (medfly)) in each sampling interval and at different distances from the release point during the three spring 2016 trials.

**Table 1 insects-10-00268-t001:** Climate data and main features of the six mark-release–recapture trials.

Release Date	Climate Data ^1^									Prevailing Direction ^2^	Released Flies (% Recapture)	
	T Mean (°C)	T Max (°C)	T Min (°C)	RH Mean (%)	RH Max (%)	RH Min (%)	Sun Rad	Wind Speed Max (m/s)	Wind Speed Mean (m/s)		*D. suzukii*	*C. capitata*
14-Oct-15	14.8	21.2	9.5	77	97	46	166	1.0	3.6	W-WN	500 (18.8)	1000 (25)
27-Oct-15	17.3	23.6	11.9	69	92	42	144	1.0	3.9	W-WN	500 (30.6)	3000 (22.5)
11-Nov-15	15.1	20.9	11.3	88	99	65	110	0.7	2.5	W	1000 (2.1)	3000 (11.8)
12-Apr-16	16.5	24.2	9.8	60	93	32	267	1.3	5.0	W	2000 (3.8)	3000 (68.7)
26-Apr-16	14.6	19.8	9.2	74	95	49	203	1.1	4.5	NE	2000 (3.1)	3000 (9.8)
17-May-16	17.8	23.5	12.5	77	96	53	274	1.0	4.2	W	2000 (2.2)	3000 (11.7)

^1^ Temperature (T), relative humidity (RH), sun radiation (Sun Rad) and wind speed. The mean, max and min values obtained during the 48 h sampling after the fly releases in October and November, and for 1 week after the releases in April and May. ^2^ Prevailing wind direction: North (N), South (S), West (W), East (E).

**Table 2 insects-10-00268-t002:** Statistics of mark-release–recapture experiments *.

Insect	Season	Orientation (O)	Distance (D)	Time (T)	O × D	D × T
*D. suzukii*	autumn	*p* = 0.913	*F*_4,345_ = 7.73 *p* < 0.001	*F*_2,345_ = 31.03 *p* < 0.001	-	*F*_8,345_ = 3.48 *p* < 0.001
	spring	*F*_7,453_ = 2.57 *p* = 0.013	*F*_4,453_ = 49.95 *p* < 0.001	*F*_3,453_ = 5.88 *p* < 0.001	-	*F*_12,453_ = 4.91 *p* < 0.001
*C. capitata*	autumn	*F*_7,318_ = 4.15 *p* < 0.001	*F*_4,318_ = 85.57 *p* < 0.001	*F*_2,318_ = 7.20 *p* < 0.001	*F*_28,318_ = 2.99 *p* < 0.001	-
	spring	*p* = 0.681	*F*_4,460_ = 12.12 *p* < 0.001	*F*_3,460_ = 1.72 *p* = 0.161	*-*	*F*_12,460_ = 2.18 *p* = 0.012

* ANOVA results (Post-hoc Tukey HSD tests, *p* < 0.05) using the log-transformed data of the total flies captured per time interval. No significant interaction Orientation × Time was obtained in any case.

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
