# Peer review of "Survey on Drosophila suzukii Natural Short-Term Dispersal Capacities Using the Mark−Release−Recapture Technique"

_insects, 2019, doi:10.3390/insects10090268_

Round 1

Reviewer 1 Report

This manuscript describes a mark-recapture study using two species: D. suzukii & C. capitata. Overall I thought this was a well-done project, that was nicely written. I have a few small questions and concerns that I hope the authors will be able to address:

In your description of your ELISA (Section 2.6), you do not mention if/how you blocked your plates. This information would be useful for those looking to use this method in the future.

In many species there is sexual dimorphism in dispersal. It would be useful if in addition to your combined-sex analyses, that you examined the factors associated with recapture for your males and females individually. This would provide valuable insight into the behavioural ecology of this species. Furthermore information on the sex ratios of the released samples would also be beneficial.

Given the differences in lures used to try and capture D. suzukii & C. capitata, you should use more caution when comparing their dispersal abilities, as some of the differences might be associated with the efficacy of your sampling methods. This is especially true with D. suzukii as there was such low recapture rates. Those that you recaptured may not be representative of the whole population (another reason to look at the sexes separately).

Author Response

Point 1: In your description of your ELISA (Section 2.6), you do not mention if/how you blocked your plates. This information would be useful for those looking to use this method in the future.

Response 1: According to the ELISA plate manufacturer, plate well blocking is not necessary provided that Tween 20 is used in the washing and assay buffer throughout the ELISA procedure as described.

Point 2: In many species there is sexual dimorphism in dispersal. It would be useful if in addition to your combined-sex analyses, that you examined the factors associated with recapture for your males and females individually. This would provide valuable insight into the behavioural ecology of this species. Furthermore information on the sex ratios of the released samples would also be beneficial.

Response 2: We totally agree with the referee that this information would be useful; unfortunately, sex was not determined during the trials and we do not have these data. 

Point 3: Given the differences in lures used to try and capture D. suzukii & C. capitata, you should use more caution when comparing their dispersal abilities, as some of the differences might be associated with the efficacy of your sampling methods. This is especially true with D. suzukii as there was such low recapture rates. Those that you recaptured may not be representative of the whole population (another reason to look at the sexes separately).

Response 3: Following your suggestion (the other two referees also agree), we have removed any comparison between D. suzukii and C. capitata and we have include the C. capitata data only to check that the trial is working and no insecticide or other indirect factors are affecting the field trial.

Reviewer 2 Report

“Survey on Drosophila suzukii natural short-term dispersal capacities using the mark-release-recapture technique”

This is an interesting study investigating the recapture of SWD in orchards using the immunomarking technique. We definitely need more information on the dispersal capacities of SWD. Overall the design is good, but the data analysis and presentation of results need significant improvement. See specific comments for each section below.

The manuscript could benefit from professional editing to improve the sentence structure and flow. I recommend the Duke University Scientific Writing Resource https://cgi.duke.edu/web/sciwriting/, and also “The Science of Scientific Writing” by Gopen and Swan.

Line 50: It might be obvious, but what is the difference between MRR and mark-capture?

Line 61: Why were the flies irradiated? If this has some relevance to SIT, it needs to be explained in the introduction.

Line 88: Unpublished results from what?

Line 124, 134: D. suzukii..”

Lines 119 – 134: How many flies were released each time? SWD pupae are particularly fragile, and it doesn’t state roughly what percentage of the 400-500 irradiated pupae actually emerged as adults.

Line 142: What was added? This is a good example of how the structure makes the sentences difficult to read some times. Again, check out the resources I recommended for scientific writing.

Line 152: What did you use as a threshold to determine what was scored as “positive”? In many immunomarking assays, a common protocol is to use three standard deviations above the mean of a negative control.

Table 1: Include number of flies released on each date

Line 160-2: A clarification here. Was this a multivariate analysis of variance (MANOVA) using two outcomes (captured SWD and captured Medflies), or was it two separate ANOVAs for each fly species?

Line 162: Was this analysis done separately for each release, or was the date of release included as a covariate or random factor? Furthermore, what is considered a replicate here for the purpose of the ANOVA? Is each of the 8 traps one replicate (therefore n = 8)?

General methods: Given the potential for spatial autocorrelation and zero-inflated data, I am not sure an analysis of variance is the most appropriate test for this type of data. I recommend reading “Case study in data analysis: Variables related to Codling moth abundance and the efficacy of the Okanagan Sterile Insect Release program” by Esterby et al. for examples on how to analyse spatial recapture data.

Results: In some places, the authors use SWD, and in others, D. suzukii. Pick one term and stick with it.

Line 167: See comment from lines 160-2. Did you originally run a MANOVA then?

Line 168: Ok, this explains my previous comment from line 162. Perhaps better to include this in methods?

Line 176: I don’t think this was a low value. If you compare the literature on immunomarking and dispersion in other taxa, you’ll see that recapture rate often ranges below 20%.

Table 2: Clarify what is “O”, “T”, and “D”. A table should be self-explanatory, without having to refer to the text.

Table 2: How were the results for each date pooled to have a total for the season? Added or averaged? Again, how is each replicate defined?

There is no need to have a O x T column if it is all blank, or explain why there are no results here.

Figure 3: Although this shows the total number of flies collected, it does not illustrate a comparison of actual recapture rate between SWD and medfly. An additional graph or secondary axis with % recapture would be useful.

Line 196: Table 2 does not show recapture rates.

Figure 4: What is “1 sem”? As a native Spanish speaker, I can assume it means “1 week”, but please format your results properly!

Line 222: Table 2 does not show recapture rates.

Line 231-32: The first sentences of the discussion are probably the most important sentences of your manuscript. What is written here has nothing to do with this study.

Line 236: This recapture rate comparison is buried and poorly presented in the results, and it forces the reader to constantly jump back and forth to different paragraphs to search for this information. I suggest having a side-by-side comparison of the % recapture rates (not totals) for SWD and Medfly, either in a graph or table.

Line 285: This is not necessarily true. Trap captures does not equate number of individuals, and as you mentioned it previously, the lure for medfly is probably more efficient than an ACV lure.

Author Response

The manuscript could benefit from professional editing to improve the sentence structure and flow. I recommend the Duke University Scientific Writing Resource https://cgi.duke.edu/web/sciwriting/, and also “The Science of Scientific Writing” by Gopen and Swan.

Response: Following your suggestion the manuscript has been edited by professional editing services.

Point 1: Line 50: It might be obvious, but what is the difference between MRR and mark-capture?

Response 2: We have explained the difference in the text  “Marking techniques are commonly used to study natural movement and distribution of insects in the field, including mark-release-recapture (MRR) where reared insects are marked, released and recaptured and mark-capture experiments in which wild insects resulted marked and their movement is studied in traps located around the marking point

Point 2: Line 61: Why were the flies irradiated? If this has some relevance to SIT, it needs to be explained in the introduction.

Response 2: “The reason to use irradiated D. suzukii is to avoid releasing a potential damaging pest population in the area”. We do not expect any relevance to SIT as the biotic potential of this species is very high to apply SIT. This conclusion was obtained after a brief discussion with Jorge Hendrichs (IAEA-FAO Insect division) about the possibility of using SIT with D suzukii 5 years ago.

Point 3: Line 88: Unpublished results from what?

Response 3: We have include the unpublished data “as this dose ensures over 99% offspring reduction”

Point 4: Line 124, 134: D. suzukii..”

Response 4: Addressed in italics

Point 5: Lines 119 – 134: How many flies were released each time? SWD pupae are particularly fragile, and it doesn’t state roughly what percentage of the 400-500 irradiated pupae actually emerged as adults.

Response 5: Reviewer was right. This data was not provided. We have included this sentence Emergence of irradiated D. suzukii at 40 Gy was studied in three irradiations and resulted in 61.4±4.7% (average± SE). C. capitata emergence at 105 Gy was 94% (± 1.7) (results provided by Local Government)”.

Point 6: Line 142: What was added? This is a good example of how the structure makes the sentences difficult to read some times. Again, check out the resources I recommended for scientific writing.

Response 6: This sentence was not clear. We have changed it to: “Next, 100 µl of fly washing buffer was added in each well and plates were incubated at room temperature for 30 min”

Point 7: Line 152: What did you use as a threshold to determine what was scored as “positive”? In many immunomarking assays, a common protocol is to use three standard deviations above the mean of a negative control.

Response 7:  We have scored as positive values of absorbance three standard deviation above or below the mean of the positive control and at least 10 times more absorbance than negative control. New sentence D. suzukii marking was considered positive if absorbance was three standard deviation above or below the mean of the positive control and at least 10 times more absorbance than negative control.”

Point 8: Table 1: Include number of flies released on each date

Response 8: As suggested, we have included the number of released flies and % of recapture for each trial in Table 1.

Point 9: Line 160-2: A clarification here. Was this a multivariate analysis of variance (MANOVA) using two outcomes (captured SWD and captured Medflies), or was it two separate ANOVAs for each fly species?

Response 9: We have modified the Statistics section and now is more clear: “Data variability of total numbers of D. suzukii flies and male medflies captured per time interval was analyzed according to, species, distance from the central release point, time passed after fly release, date of release and trap orientation, using multifactor analysis of variance (MANOVA). Posterior ANOVA were conducted for each specie and season.”

Point 10: Line 162: Was this analysis done separately for each release, or was the date of release included as a covariate or random factor? Furthermore, what is considered a replicate here for the purpose of the ANOVA? Is each of the 8 traps one replicate (therefore n = 8)?

Response 10: Date was included as a random factor in the analysis as now it is stated in response 9. Orientation was considered as a factor and it was included in the analysis so we considered 3 replicates in 3 dates.

Point 11: General methods: Given the potential for spatial autocorrelation and zero-inflated data, I am not sure an analysis of variance is the most appropriate test for this type of data. I recommend reading “Case study in data analysis: Variables related to Codling moth abundance and the efficacy of the Okanagan Sterile Insect Release program” by Esterby et al. for examples on how to analyse spatial recapture data.

Response 11: We have read the paper recommended. I am sorry but we can not apply this model to our study in the revision time as we should contact a statistical professor to conduct this work and the mathematical model is very complex. We have use the same methodology that “Todd E. Shelly and James Edu “Mark-release-recapture of males of Bactrocera cucurbitae and B. dorsalis (Diptera: Tephritidae) in two residential areas of Honolulu” in Journal of Asia-Pacific Entomology 2010 (13: 131-137)”. We have included this reference in the text

Point 12: Results: In some places, the authors use SWD, and in others, D. suzukii. Pick one term and stick with it.

Response12: We have removed SWD from the text and Mediterranean fruit fly has been changed to C. capitata

Point 13: Line 167: See comment from lines 160-2. Did you originally run a MANOVA then?

Response 13: Yes, it is now included in the Data analysis section

Point 14: Line 168: Ok, this explains my previous comment from line 162. Perhaps better to include this in methods?

Response 14: Ok. Included in Material and methods section (Data analysis)

Point 15: Line 176: I don’t think this was a low value. If you compare the literature on immunomarking and dispersion in other taxa, you’ll see that recapture rate often ranges below 20%.

Response 15: We agree and we have changed the number of the table according to Point 8. Now the sentence is “The mean number of released D. suzukii was 667 flies and percentage of recapture was showed in Table 1

Point 16: Table 2: Clarify what is “O”, “T”, and “D”. A table should be self-explanatory, without having to refer to the text.

Response 16: for clarity purposes, now Orientation, Distance and Time are written in capital letters to clearly identify the meaning of the initials.

Point 17: Table 2: How were the results for each date pooled to have a total for the season? Added or averaged? Again, how is each replicate defined?

Response 17: For the statistical analysis each data obtained was included in the ANOVA: we have include a note in the table: three replications per treatment were made in each season. As season is a significant factor, each season was analyzed individually.

Point 18: There is no need to have a O x T column if it is all blank, or explain why there are no results here.

Response 18: We have removed the column and added a new note below the table “No significant interaction orientation x time is obtained in any case

Point 19: Figure 3: Although this shows the total number of flies collected, it does not illustrate a comparison of actual recapture rate between SWD and medfly. An additional graph or secondary axis with % recapture would be useful.

Response 19: we have included this data in table 1. As we have removed any comparison between dispersal capacity of SWD and medfly following reviewers suggestion we have not include this data again in Figure 3.

Point 20: Line 196: Table 2 does not show recapture rates.

Response 20: Ok. Changed to table 1 according to point 8

Point 21: Figure 4: What is “1 sem”? As a native Spanish speaker, I can assume it means “1 week”, but please format your results properly!

Response 21: Sorry for the typo. Changed

Point 22: Line 222: Table 2 does not show recapture rates.

Response 22: Changed to Table 1 According to point 8

Point 23: Line 231-32: The first sentences of the discussion are probably the most important sentences of your manuscript. What is written here has nothing to do with this study.

Response 23: Agree that this is not the most important part of the study. We have restructure the discussion section following your suggestion.

Point 24: Line 236: This recapture rate comparison is buried and poorly presented in the results, and it forces the reader to constantly jump back and forth to different paragraphs to search for this information. I suggest having a side-by-side comparison of the % recapture rates (not totals) for SWD and Medfly, either in a graph or table.

Response 24: Recapture rate is now in table 1.

Point 25: Line 285: This is not necessarily true. Trap captures does not equate number of individuals, and as you mentioned it previously, the lure for medfly is probably more efficient than an ACV lure.

Response 25: Totally agree. We have removed any comparison between D. suzukii and C .capitata in the text following your suggestion. The new sentence are as follow: “As a conclusion, D. suzukii has a low dispersal capacity when compared with reported distances for other fruit flies [46]. We can estimate a dispersal distance for D. suzukii between 200 and 300 m when no prevailing wind takes place. This conclusion agrees with previous studies in which high activity levels of D. suzukii takes place in the field margin of cultivated crops surrounded by host plants [47].”

Reviewer 3 Report

This manuscript details the attempt to track and compare mark release recapture methods for two fruit infesting flies, Ceratitis capitata and Drosophila suzukii. Over the course of two seasons, flies of each species are marked with egg protein, released at six time points, and traps placed at set distances were checked at regular intervals for flies. Differences between the two fly behaviors were noted in terms of distances flown, average time to recapture and seasonality. 

While certain results presented are interesting, such as why differences between spring and fall releases were observed, having only one year of data, and only one release location make any result hard to interpret. 

Furthermore, while these are two fruit infesting flies, their biology, evolution, behavior, and type of attractant available do not make them good comparators. The two different collection methods, while acknowledged, are indeed monitoring two different phenomena – namely food vs mate seeking behavior. And while a sex pheromone may be useful most times of the year, food-based attractants only attract certain types of D. suzukii flies, the number and proportion of which differ throughout the year [see Swoboda-Bhattarai, K.A., McPhie, D.R. and Burrack, H.J., 2017. Reproductive status of Drosophila suzukii (Diptera: Drosophilidae) females influences attraction to fermentation-based baits and ripe fruits. Journal of economic entomology110(4), pp.1648-1652.] Any comparison between the flies needs to fully acknowledge the large differences among them. Given the different collection methods and the evolutionary distance between the flies, it does not seem like enough connects them to justify the comparisons made. Stated objective of study was to look at short-term dispersal of D. suzukii(Line 60). A careful study of this would be interesting in and of itself. 

The overall method of immunomarking flies is fine, but again D. suzukii was protein marked while C. capitata were dusted. More information and consideration needs to be given to the number of flies released, what specific hosts are available, their density, and where. While alluded to, this is not detailed or taken into account in the analysis. Host density and location may be just as, or more important than wind direction with regard to trap orientation results. These are both highly polyphagous flies with some host overlap, but likely a different preference hierarchy. 

Overall suggestion to improve this manuscript would be to add additional release years and locations of releases, better details on the hosts in the area. Also I suggest considering the biology of the fly during the release dates. Is there a reason to think spring and fall would be the same or different for fly dispersal behavior given the temperature and day length effects on D. suzukii physiology? How is this taken into account for in your methods using laboratory flies?

Minor suggestions:

- Inconsistent language usage, switches back and forth between SWD and D. suzukii. Stick to one, as this is not done with C. capitata.

- Be sure to check that Latin names (e.g., D. suzukii) are always italicized. 

- L106 – how do you know there was no interaction between the two attractants used as they were in the same headspace? It seems likely, and would be good to check if there’s any interference or synergy, in the case of C. capitata as the males were likely also attracted to the food bait.

- L144 – a better negative control would be to wash an unmarked, sterilized fly

- L176-7- Table 2 does not show % recapture

- Figure 4 – Change 1sem to English term, either in hours or days

- L231 – Better to say D. melanogaster 

- L253 – ripen -> ripe

- L257 – delete ‘yet’

- L258 – genus should be genera

- L287 – stablished -> established

Author Response

Point 1: Furthermore, while these are two fruit infesting flies, their biology, evolution, behavior, and type of attractant available do not make them good comparators. The two different collection methods, while acknowledged, are indeed monitoring two different phenomena – namely food vs mate seeking behavior. And while a sex pheromone may be useful most times of the year, food-based attractants only attract certain types of D. suzukii flies, the number and proportion of which differ throughout the year [see Swoboda-Bhattarai, K.A., McPhie, D.R. and Burrack, H.J., 2017. Reproductive status of Drosophila suzukii (Diptera: Drosophilidae) females influences attraction to fermentation-based baits and ripe fruits. Journal of economic entomology110(4), pp.1648-1652.] Any comparison between the flies needs to fully acknowledge the large differences among them. Given the different collection methods and the evolutionary distance between the flies, it does not seem like enough connects them to justify the comparisons made. Stated objective of study was to look at short-term dispersal of D. suzukii(Line 60). A careful study of this would be interesting in and of itself.  

Response 1: We have included these considerations in the discussion section and we have removed the comparisons between the two flies according to reviewer 1 and 3 suggestions. 

Point 2: The overall method of immunomarking flies is fine, but again D. suzukii was protein marked while C. capitata were dusted. More information and consideration needs to be given to the number of flies released, what specific hosts are available, their density, and where. While alluded to, this is not detailed or taken into account in the analysis. Host density and location may be just as, or more important than wind direction with regard to trap orientation results. These are both highly polyphagous flies with some host overlap, but likely a different preference hierarchy. 

Response 2: We have included more information about the released flies in Table 1 and details about the emergence of each species. Regarding hosts, we selected this experimental field because no other hosts or crops were available in a distance over 1 km. We have included in the text the following information: “About host phenology, trial fields are very homogeneus with no other host present in less than 1.3 km around the release point except mid-season clementines. We found blackberries in a ravine in the south-east direction at 1.3 km and some backyards 1.4 km away in the east direction. However no influence of these crops has been observed in D. suzukii movement”.

Point 3: Overall suggestion to improve this manuscript would be to add additional release years and locations of releases, better details on the hosts in the area. Also I suggest considering the biology of the fly during the release dates. Is there a reason to think spring and fall would be the same or different for fly dispersal behavior given the temperature and day length effects on D. suzukii physiology? How is this taken into account for in your methods using laboratory flies?

Response 3: Sorry, but we are not able to repeat the trials now, as the project finished one year ago. Nevertheless, we would like to highlight that we have conducted 6 release-recapture experiments, in two different seasons, to obtain the conclusions and we think that this might be enough to provide valuable information about the movement of the flies. We agree that repeating the same trials during several years in different crops would be more powerful for the conclusions, but it is not affordable at the moment. I guess that conclusion about the movement of D suzukii depending on the season should be confirmed with more replicates, so we have moderated our conclusion about this point.

Minor suggestions:

Point 4: Inconsistent language usage, switches back and forth between SWD and D. suzukii. Stick to one, as this is not done with C. capitata.

Response 4: We have removed all SWD and Mediterranean fruit flies from the text (only in tables due to space available)

Point 5: Be sure to check that Latin names (e.g.,D. suzukii) are always italicized.

Response 5: Done  

Point 6: L106 – how do you know there was no interaction between the two attractants used as they were in the same headspace? It seems likely, and would be good to check if there’s any interference or synergy, in the case of C. capitata as the males were likely also attracted to the food bait.

Response 6: We previously placed D. suzukii traps in citrus orchards and Ceratitis capitata captures were very low (1 or 2 flies per week) meanwhile trimedlure baited traps caught more than 50 C. capiata per week and no D. suzukii were detected. So we do not expect any interference.

Point 7: - L144 – a better negative control would be to wash an unmarked, sterilized fly

Response 7: We agree with this comment. In the ELISA procedure we forgot to define what were considered reference samples. Concerning negative controls, we have assayed several flies without marking on different days and ELISA results have been very similar to the ones obtained just with buffer as negative control in every plate. Therefore, in the revised manuscript we have added (section 2.6) how was a fly scored as positive, that is, when the absorbance values were higher than the mean of negative control (n=10, non-marked flies) plus four standard deviations. We have included this sentence in the text: “A threshold was established as the mean of negative samples (10 values of different plates) plus four standard deviations. Wells showing absorbance above this threshold were considered as positive, that is, contained a marked fly

Point 8- L176-7- Table 2 does not show % recapture

Response 8: Now it is included in table 1. Changed in the text

Point 9- Figure 4 – Change 1sem to English term, either in hours or days

Response 9: Done

Point 10 L231 – Better to say D. melanogaster 

Response 10: Changed as suggested

Point 11- L253 – ripen -> ripe

Response 11: Changed as suggested

Point 12- L257 – delete ‘yet’

Response 12: Deleted

Point 13- L258 – genus should be genera

Response 13: Changed as suggested

Point 14- L287 – stablished -> established

Response 14: done

Reviewer 4 Report

See attached file.

Author Response

Point 1: My suggestion is to include a bit about this result so that the underlying setup for the important part of the experimental design can be visualized.

Reply: We have included the following text "In a preliminary experiment flies were irradiated with 10 and 40 Gy and they were allowed to mate with non-irradiated ones. As a result of the 40 Gy dose, females produced few eggs, only 3.6% of those hatched and 0.8% finished their development to pupae. The 10 Gy dose had less effect; egg hatching was reduced only to 27.7% and 43.1% of the emerged larvae still reached pupal stage"  

Point 2: Only a few minor grammatical errors  

Reply: We have revised the text and addressed the detected errors

Round 2

Reviewer 1 Report

The authors have addressed the issues I raised sufficiently. It is a shame that sex data was not collected, as this missed a great opportunity to get insight into the biology of this species .

Author Response

Point 1: The authors have addressed the issues I raised sufficiently. It is a shame that sex data was not collected, as this missed a great opportunity to get insight into the biology of this species .

Thank you for your suggestion. We will take this point into account for the next study. 

Reviewer 2 Report

The narrative of the manuscript was significantly improved and it is more clear to follow.

The statistical analysis has concerning flaws. Spatial data is rarely independent, therefore it rarely meets the assumptions of an analysis of variance. There is no test for spatial autocorrelation or corrections for zero-inflated data. The authors followed the methods from another paper, but it does not necessarily make it correct, nor does a short deadline for resubmission.

The results of the MANOVA are not shown to support a post-hoc ANOVA. In the reply they state that there are now 3 replicates in 3 dates, I assume they mean 3 replicates (as in 3 dates) in each season. For the assumptions of an analysis of variance, these are not independent replicates, these are repeated measures.

Author Response

Point 1: The results of the MANOVA are not shown to support a post-hoc ANOVA. In the reply they state that there are now 3 replicates in 3 dates, I assume they mean 3 replicates (as in 3 dates) in each season.

Reply: Yes, we have conducted the trials three times each season.

Point 2: For the assumptions of an analysis of variance, these are not independent replicates, these are repeated measures.

Reply: We do not agree with this consideration. We are not using repeated measures within the same trial for the statistical analysis. The three trials in each season are independent. For example: the captures of the second release do not depend on the flies of the first release. For each release we have used different irradiated flies and we have released the flies six different times (three per season).